# Antiproliferative Effect of Acridine Chalcone Is Mediated by Induction of Oxidative Stress

**DOI:** 10.3390/biom10020345

**Published:** 2020-02-22

**Authors:** Peter Takac, Martin Kello, Maria Vilkova, Janka Vaskova, Radka Michalkova, Gabriela Mojzisova, Jan Mojzis

**Affiliations:** 1Department of Pharmacology, Faculty of Medicine, Pavol Jozef Safarik University, 040 11 Kosice, Slovakiaradka.michalkova@student.upjs.sk (R.M.); 2Institute of Human and Clinical Pharmacology, University of Veterinary Medicine and Pharmacy, 041 81 Kosice, Slovakia; 3Department of Organic Chemistry, Faculty of Science, Pavol Jozef Safarik University, 040 01 Kosice, Slovakia; maria.vilkova@upjs.sk; 4Department of Medical and Clinical Biochemistry, Faculty of Medicine, Pavol Jozef Safarik University, 040 01 Kosice, Slovakia; janka.vaskova@upjs.sk; 5Department of Experimental Medicine, Faculty of Medicine, Pavol Jozef Safarik University, 040 01 Kosice, Slovakia; gabriela.mojzisova@upjs.sk

**Keywords:** chalcones, antiproliferative, oxidative stress

## Abstract

Chalcones are naturally occurring phytochemicals with diverse biological activities including antioxidant, antiproliferative, and anticancer effects. Some studies indicate that the antiproliferative effect of chalcones may be associated with their pro-oxidant effect. In the present study, we evaluated contribution of oxidative stress in the antiproliferative effect of acridine chalcone 1C ((2 E)-3-(acridin-9-yl)-1-(2,6-dimethoxyphenyl)prop-2-en-1-one) in human colorectal HCT116 cells. We demonstrated that chalcone 1C induced oxidative stress via increased reactive oxygen/nitrogen species (ROS/RNS) and superoxide production with a simultaneous weak adaptive activation of the cellular antioxidant defence mechanism. Furthermore, we also showed chalcone-induced mitochondrial dysfunction, DNA damage, and apoptosis induction. Moreover, activation of mitogen activated phosphokinase (MAPK) signalling pathway in 1C-treated cancer cells was also observed. On the other hand, co-treatment of cells with strong antioxidant, *N*-acetyl cysteine (NAC), significantly attenuated all of the above-mentioned effects of chalcone 1C, that is, decreased oxidant production, prevent mitochondrial dysfunction, DNA damage, and induction of apoptosis, as well as partially preventing the activation of MAPK signalling. Taken together, we documented the role of ROS in the antiproliferative/pro-apoptotic effects of acridine chalcone 1C. Moreover, these data suggest that this chalcone may be useful as a promising anti-cancer agent for treating colon cancer.

## 1. Introduction

Chalcone, (E)-1,3-diphenyl-2-propene-1-one, has been reported as being a precursor in flavonoid synthesis [1]. Similar to flavonoids, chalcones have already been documented as exhibiting several biological activities, including antioxidant [2], anti-inflammatory [3], antibacterial and antifungal [4], antiprotozooal [5], immunomodulatory [6], and antiangiogenic [7,8] effects. Furthermore, antiproliferative/antitumor activities of chalcones have also been intensively studied [9,10,11,12,13,14]. The available studies revealed multitargeted activity of chalcones including various kinases [15], microtubules [16], multidrug-resistance proteins [17], or different signalling pathways associated with cell survival or death [18,19]. Moreover, we and others have detected the ability of chalcones to inhibit cancer cell proliferation via cell cycle arrest [20,21,22,23].

As was mentioned above, chalcones possesses antioxidant properties. Oxidative stress can be involved in cellular dysfunction due to DNA, protein, or lipid damage [24]. Oxidative cell damage can be linked to diseases such as cancer or neurodegenerative, cardiovascular, or metabolic diseases [25,26,27] and chalcones are suggested to either prevent or slow progression of these chronic diseases [28,29,30].

On the other hand, chalcones, under certain circumstances, may act as oxidants [31], and this effect can be associated with their antitumor activity [32,33]. The pro-oxidant activity of chalcones may result from different mechanisms such as increase in superoxide formation [34], cellular glutathione (GSH) depletion [35], or generation of phenoxyl radicals [36].

In our previous paper, we documented the ability of acridine chalcone 1C ((2 E)-3-(acridin-9-yl)-1-(2,6-dimethoxyphenyl)prop-2-en-1-one) to suppress growth of cancer cells in vitro [11]. In accordance with the above-mentioned pro-oxidative effect of chalcones, we hypothesized that antiproliferative effect of chalcone 1C can be associated with either reactive oxygen species (ROS) or reactive nitrogen species (RNS) formation. To verify our hypothesis, we aimed to elucidate the link between ROS/RNS and the antiproliferative effect of 1C in human colorectal carcinoma cells HCT116. 

The results of our study indicated that antiproliferative and pro-apoptotic effects of 1C are free radical-dependent, and these effects were inhibited by a powerful antioxidant *N*-acetylcysteine (NAC).

## 2. Materials and Methods 

### 2.1. Tested Compound

(2 E)-3-(Acridin-9-yl)-1-(2,6-dimethoxyphenyl)prop-2-en-1-one (1C) was synthetized by Maria Vilkova (Faculty of Natural Sciences of the P.J. Šafarik University, Košice). The structure of compounds was confirmed by using ^1^H, ^13^C nuclear magnetic resonance (NMR), infrared (IR) spectroscopy, and mass spectrometry. The studied agent was dissolved in DMSO. The final concentration of DMSO in the culture medium was 0.02% and exhibited no cytotoxicity. 

### 2.2. Cell Culture

Cell line HCT116 (human colorectal carcinoma) was cultured in RPMI 1640 medium (Biosera, Kansas City, MO, USA). The growth medium was supplemented with 10% foetal bovine serum and 1X HyClone antibiotic/antimycotic solution (GE Healthcare, Little Chalfont, United Kingdom). Cells were cultured in an atmosphere containing 5% CO_2_ in humidified air at 37 °C. Cell viability, estimated by trypan blue exclusion, was greater than 95% before each experiment.

### 2.3. Viability Test

The MTT (3-(4,5-dimethylthiazol-2-yl)-2,5-diphenyltetrazolium bromide)colorimetric test was used to determine the antiproliferative effect of 1C and *N*-acetylcysteine. HCT116 cells (5 × 10^3^/well) were seeded in 96-well polystyrene microplates (SARSTEDT, Nümbrecht, Germany). Twenty-four hours after seeding, 1C final concentration (10 µM) and NAC (final concentration 0.3, 0.5, 1, 1.5, 2, and 2.5 mM) or their combinations were added. After 72 h, cells were incubated with 10 μL of MTT (5 mg/mL, Sigma-Aldrich Chemie, Steinheim, Germany) at 37 °C. After an additional 4 h, insoluble formazan produced by metabolic reactions were dissolved by 100 μL of a 10% sodium dodecyl sulphate. Cell proliferation was evaluated by measuring the absorbance at wavelength 570 nm using the automated Cytation 3 Cell Imaging Multi-Mode Reader (Biotek, Winooski, VT, USA). Absorbance of control wells was taken as 1.0 = 100%, and the results were expressed as a fold/percentage of untreated control.

### 2.4. Flow Cytometry Analyses

#### 2.4.1. Analysis of Cell Cycle

For flow cytometric analysis of the cell cycle, floating and adherent HCT116 cells were harvested 24, 48, and 72 h after treatment (1C, 10 µM), then washed in cold phosphate-buffered saline (PBS), fixed in cold 70% ethanol, and kept at −20 °C overnight. Before each analysis, cells were washed in PBS, resuspended in staining solution (final concentration 0.2% Triton X-100, 0.5 mg/mL ribonuclease A and 0.025 mg/mL propidium iodide in 500 µL PBS (all Sigma Aldrich, St. Louis, MO, USA)), and incubated for 30 min in the dark at room temperature. The DNA content of stained cells was analysed using a FACSCalibur flow cytometer (Becton Dickinson, San Jose, CA, USA)). 

#### 2.4.2. Flow Cytometry Analysis of Free Radicals, Apoptosis, Signalling Pathways, and DNA Damage

HCT116 cells were seeded in Petri dishes and cultivated for 24 h in a complete medium with 10% FBS. Cells were treated with 1C (10 µM) or with *N*-acetylcysteine (2.5 mM) and their mutual combination at 1, 3, 6, 24, 48, and 72 h prior to analysis. The antioxidant NAC was used in mutual combination as pre-treatment for 30 min before 1C was added. Floating and adherent cells were harvested, washed in PBS, distributed for particular analysis, and stained before analysis (see Table 1). After 15 min incubation at room temperature in the dark, samples were put on ice and fluorescence changes were detected by a flow cytometer BD FACSCalibur (BD Biosciences). A minimum of 1 × 10^4^ events were analysed per analysis.

### 2.5. Antioxidant Enzyme Activities and Glutathione Content Measurement

Both floating and adherent HCT116 cells were harvested at 1, 3, 6, 24, 48, and 72 h after treatment with 1C (10 µM), NAC (2.5 mM), or their mutual combination. The activity of glutathione reductase (GR; EC 1.6.4.2) was measured according to a modified method described by Carlberg and Mannervik [37]. Activity of glutathione peroxidase (GPx, EC 1.11.1.9) was measured according to Zagrodski et al. [38]. For detection of glutathione-S-transferase (GST, EC 2.5.1.18) activity, Glutathione S-Transferase Assay Kit (Sigma-Aldrich, Germany) was used. Reduced glutathione (GSH) content was measured by the method originally described by Floreani et al. [39]. Assays were performed on an M 501 single beam UV/VIS spectrophotometer (Spectronic Camspec Ltd., Leeds, United Kingdom). All measured parameters were calculated per milligram or gram of protein determined using the bicinchoninic acid assay.

### 2.6. Western Blot Analysis

HCT116 cells were treated with compound 1C (10 µM), NAC (2.5 mM), or their mutual combinations for 24, 48, and 72 h. Protein lysates from HCT116 cells were prepared using a lysis buffer containing 1 mol/L Tris/HCl (pH 6.8), glycerol, 20% SDS (sodium dodecyl sulphate), and deionized H_2_O in the presence of PIC (protease inhibitor cocktail) and a process of sonication. The concentration of proteins was determined using the Pierce BCA Protein Assay Kit (Thermo Scientific, Rockford, IL, United States) and measured using an automated Cytation 3 Cell Imaging Multi-Mode Reader (Biotek) at wavelength 570 nm. Forty micrograms of total cellular proteins were separated on SDS-PAA gel (12%) at 100 V for 2 h and electrotransferred onto PVDF Blotting Membrane (GE Healthcare, Chicago, IL, United States) at 200 mA for 2 h using a BioRad Mini Trans-Blot cell (BioRad, Hercules, CA, USA). The membrane was then blocked in 4% milk with TBS-Tween (pH 7.4) for 1 h at room temperature to minimize non-specific binding. After that, the membrane was incubated with primary antibodies overnight at 4 °C. Immunoblotting was carried out with the antibodies stated below (Table 2). After incubation with primary antibodies, membranes were washed in TBS-Tween (3 × 5 min) and incubated with corresponding horseradish peroxidase-conjugated secondary antibodies for 1 h at room temperature. After incubation and washing of membranes (3 × 5 min with TBS-Tween) the expression of selected proteins was detected by chemiluminescent ECL substrate (Thermo Scientific, Rockford, IL, USA) and MF-ChemiBIS 2.0 Imaging System (DNR Bio-Imaging Systems, Jerusalem, Israel).

### 2.7. Statistical Analysis

Results are expressed as mean ± standard deviation (SD). Statistical analysis of the data was performed using standard procedures, with one-way ANOVA followed by the Bonferroni multiple comparisons test. Values of *p* < 0.05 were considered as being statistically significant.

## 3. Results

### 3.1. Viability of HCT116 Cells after 1C and NAC Treatment

To verify our hypothesis that the antiproliferative effect of 1C could be associated with free radical production, we analysed viability/proliferation of HCT116 cells exposed to 1C (10 µM) alone or in combination with NAC (0.3 mM, 0.5 mM, 1 mM, 1.5 mM, 2 mM, 2.5 mM). After 72 h of incubation, 1C significantly decreased HCT116 cell viability. However, when combined with NAC, the effect of 1C on cell proliferation was significantly attenuated. A significant protective effect of NAC was observed at the concentration range of 1.0–2.5 mM (Figure 1). These results suggest that NAC exhibits an antagonistic effect on 1C-induced decrease in cell viability. Moreover, NAC in used concentrations had no inhibitory effect on HCT116 cell proliferation (Figure 1B). For further experiments, 2.5 mM concentration of NAC was selected as non-toxic.

### 3.2. NAC and 1C-Induced Oxidative Stress

In order to verify the abovementioned hypothesis that the cytotoxic effect of 1C in HCT116 cells could be related to oxidative stress, we performed several analyses focused upon free radical production or antioxidant system activity. 

The results presented in Figure 2A and Appendix A show that ROS started to be increased from 6 h of treatment (*p* < 0.05) when compared to control (untreated cells). This trend continued after 24 h, 48 h, and 72 h of treatment (*p* < 0.05; *p* < 0.001). Opposite to ROS, we observed a moderate decrease in RNS (Figure 2B and Appendix A) production after 6 h of incubation (*p* < 0.05). On the other hand, significant increase in RNS production after 24 h, 48 h, and 72 h of treatment was observed (*p* < 0.01; *p* < 0.001). In order to find out how superoxide may contribute to 1C-induced cytotoxicity, we performed a direct measurement of superoxide anion levels after treatment with 1C (Figure 2C and Appendix A). Except in the first hour of incubation, treatment of HCT116 cells with 1C significantly increased production of superoxide (*p* < 0.05; *p* < 0.01 vs. control). In both cases, co-treatment of cells with NAC significantly decreased either ROS or superoxide production (*p* < 0.05; *p* < 0.01; or *p* < 0.001 vs. 1C treated cells).

Some of the well-known consequences of free radical generation are peroxidation of polyunsaturated fatty acids and DNA damage. As our results showed (Figure 2D and Appendix A), treatment of HCT116 cells with 1C led to a significant increase in lipid peroxide level after 24 h, 48 h, and 72 h of incubation when compared to the control (*p* < 0.05; *p* < 0.01). No significant increase in lipoperoxide level was observed after shorter times of incubation (1, 3, and 6 h; Appendix A).

8-Oxo-7,8-dihydroguanine (8-oxoG) is the main product of oxidative DNA damage. Because of the clear correlation between free radical production and 8-oxoG creation, it is a frequently used cellular marker of oxidative stress. As shown in Figure 3, 1C significantly increased levels of 8-oxoG in all treatment periods (24, 48, and 72 h; *p* < 0.001). Compared with 1C treatment alone, NAC co-treatment at a dose of 2.5 mM significantly reduced some parameters of oxidative stress. Production of ROS and RNS was significantly decreased in 1C/NAC-treated after 24, 48, and 72 h of incubation (*p* < 0.05; *p* < 0.01). Similarly, significant reduction in lipid peroxides was observed in NAC-co-treated HCT116 cells (*p* < 0.01; *p* < 0.001). NAC also attenuated oxidative DNA damage. After 2.5 mM NAC addition, 8-oxoG level significantly decreased (*p* < 0.001) as compared with 1C alone treatment.

Furthermore, because oxidative stress is the result of pro-oxidants/antioxidants disbalance, we also studied effect of 1C on endogenous antioxidant status. Reduced glutathione and GSH-related enzymes play an important role in protection against reactive species produced during oxidative stress. As presented in Figure 4A and Appendix A, the content of GSH in 1C-treated cells was biphasic, with the highest level after 6 and 48 h of treatment. In addition, decrease in GSH content was detected after 24 h of incubation. Activity of GSH-related enzyme was time-dependent. Treatment of HCT116 cells with 1C resulted in increased activity of GPx and GR after 24 h of incubation only. On the other hand, activity of GST was increased during the whole course (Figure 4B–D; Appendix A). NAC modulated GSH content as well as activity of GSH-related enzymes. The content of GSH in 1C/NAC-treated cells was increased after 6 and 48 h of treatment and decreased after 24 h of treatment. This biphasic response was also observed in 1C-only-treated cells, however, combination with NAC increased content of GSH in a more appreciable manner. Moreover, NAC increased activity of GPx, GR, and GST when compared to 1C-treated cells.

### 3.3. Effect of NAC on Cell Cycle

We evaluated the cell cycle distribution of chalcone 1C-treated cells by flow cytometry. As shown in Figure 5 and Table 3, 1C induced a significant cell cycle arrest at G_2_/M phase after 24 h (*p* < 0.001) of incubation, and cell cycle progression was also arrested after 48 and 72 h of incubation (*p* < 0.01). Moreover, we also found a significant increase in number of cells with sub-G_0_/G_1_ DNA content, which is considered a marker of apoptosis (*p* < 0.01). Compared with 1C treatment alone, NAC (2.5 mM) co-treatment completely prevented G_2_/M cell cycle arrest. In addition, NAC also significantly reduced number of cells with sub-G_0_/G_1_ DNA content.

Overall, results of cell cycle analysis suggested that ROS production participated in G2/M phase arrest.

Because the analysis of the cell cycle showed the participation of free radicals in 1C-induced apoptosis, we provided a set of following experiments to study the detailed role of ROS (free radicals) in cell death processes.

### 3.4. Effect of NAC on the Presence of Apoptotic Markers

Execution of apoptosis results in the consecutive display of different biochemical markers including phosphatidylserine externalization, release of pro-apoptotic proteins, and caspase activation.

#### 3.4.1. Phosphatidylserine Externalization 

To explain the association between oxidative stress and 1C-induced cell death, double annexin V/PI (propidium iodide) staining was used, and measurements were conducted after incubation of HCT116 cells with 1C, NAC, or both. As depicted in Figure 6 and Table 4, 1C significantly reduced the percentage of living cells after 24 h of incubation, and this effect was noticed also after 48 and 72 h incubation (*p* < 0.001). Furthermore, the increase in the population of apoptotic and dead cells was also observed. However, compared with that in groups treated with 1C alone, the number of living cells significantly increased with the concomitant decrease of apoptotic and dead cells after co-treatment with NAC (*p* < 0.05, *p* < 0.01, and *p* < 0.001).

#### 3.4.2. Cytochrome *c* and Smac/DIABLO Release 

One of key apoptotic events is the release of cytochrome *c* and Smac/DIABLO from mitochondria to cytosol. Incubation of HCT116 cells with 1C led to a significant release of both cytochrome *c* and Smac/DIABLO protein after 24, 48, and 72 h of treatment. Release of both proteins was significantly attenuated by NAC co-treatment in all time periods studied (*p* < 0.05, *p* < 0.01, and *p* < 0.001) indicated role of free radicals in mitochondrial membrane permeabilization and apoptosis induction (Figure 7A,B).

#### 3.4.3. Caspase-3 and -7 Activation 

After releasing into the cytosol, cytochrome *c* forms apoptosome with subsequent activation of caspase-3 and -7 (executioner caspases), which mediate many features of apoptosis. As we found, treatment of HCT116 cells with 1C led to significant activation of caspase-3 in all time periods (*p* < 0.01; *p* < 0.001). On the other hand, significant suppression of caspase-3 activation was seen in 1C/NAC co-treated cells (*p* < 0.5 and *p* < 0.01 vs. cells treated with 1C alone) (Figure 7C). Moreover, Western blot analysis also showed that 1C increased levels of cleaved (activated) caspase-7, mainly after 48 and 72 h treatment, and NAC was able to abolish this effect (Figure 7D). 

#### 3.4.4. PARP Cleavage 

The activation of caspase-3 and -7 has been confirmed by the cleavage of PARP (substrate for caspases) using Western blot analysis. The level of cleaved PARP in 1C-treated HCT116 cells increased (48 and 72 h of incubation) as shown in Figure 7D, indicating the activation of caspase-dependent apoptosis. This analysis also revealed that the levels of cleaved PARP were markedly suppressed in NAC-co-treated HCT116 cells.

### 3.5. Effect of NAC on 1C-Induced Mitochondrial Dysfunction, Bcl-2 Phosphorylation, and DNA Damage

#### 3.5.1. Mitochondrial Membrane Potential 

It is well known that mitochondria are involved in the regulation of various functions related to cell survival or cell death. Mitochondrial membrane potential plays crucial role in mitochondria homeostasis, and long-lasting drop of it may be associated with apoptosis. As shown in Figure 8, we observed a significant decrease of MMP in chalcone 1C-treated HCT116 cells (*p* < 0.01; *p* < 0.001). By contrast, 1C/NAC co-treatment completely rescued HCT116 cells from mitochondrial membrane potential collapse (*p* < 0.01, *p* < 0.001 vs. 1C alone).

#### 3.5.2. Bcl-2 Phosphorylation

The function of anti-apoptotic Bcl-2 protein depends on phosphorylation status, and Bcl-2 phosphorylation suppresses its anti-apoptotic activity. Treatment of HCT116 cells with 1C led to a significant increase in phosphorylation of Bcl-2, thus enabling the initiation of intrinsic apoptotic cascade. In contrast to 1C, addition of NAC completely abolished Bcl-2 phosphorylation, thus preventing further apoptotic events (*p* < 0.001 vs. 1C alone) (Figure 9).

#### 3.5.3. DNA Damage 

Xenobiotic-induced DNA damage often leads to activation of DNA repair machinery including histone H2A.X, ATM (ataxia telangiectasia mutated kinase), and SMC1 protein (structural maintenance of chromosomes 1). Our results showed activation (i.e., phosphorylation) of all of the aforementioned markers of DNA damage (*p* < 0.01; *p* < 0.001) in 1C-treated cells (Figure 10A–D). Similarly to our previous experiments, co-treatment of HCT116 cells with 1C/NAC significantly prevented phosphorylation of all proteins (*p* < 0.05, *p* < 0.01, and *p* < 0.001 vs. 1C alone). 

### 3.6. Phosphorylation of p38 MAPK, JNK, and ERK1/2 Was Reduced by NAC

Because mitogen-activated protein kinase (MAPK) pathway plays an important role in apoptosis regulation, we investigated the effect of 1C on the MAPK pathway proteins, including p38 MAPK, JNK, and ERK1/2 (Extracellular signal-regulated kinase 1/2).

Findings of flow cytometry as well as Western blot analysis revealed that treatment of HCT116 cells with 1C induced an increase of the phosphorylated form of MAPK proteins at specific times (mainly after 48 and 72 h of incubation) compared to control (Figure 11A–D). However, partial reduction of MAPK protein phosphorylation was detected when NAC was used. This effect was significant after 48 and 72 h of treatment (*p* < 0.01 and *p* < 0.001 vs. 1C alone). 

## 4. Discussion

Although free radicals are involved in several physiological functions [40], uncontrolled ROS production in cells can lead to oxidative stress followed by damage of biomacromolecules, including DNA, proteins, or lipids, resulting in cell death [41]. Oxidative stress has also been linked to several diseases such as diabetes mellitus, neurodegenerative and cardiovascular diseases, and cancer [42]. Besides the endogenous antioxidant system, exogenous antioxidants, mainly phytochemicals, play an important role in the protection of cells and tissues against oxidative stress [43]. Among phytochemicals, polyphenols have been intensively studied in recent years due to their antioxidant properties [44,45]. However, in recent years, some studies have documented pro-oxidant activity of several antioxidants including polyphenols [46,47]. As was mentioned above, increased production of ROS can lead to oxidative stress and even to cell death. It should be noted that cancer cells, due to higher concentration of some ions and greater metabolic activity [48,49], are more susceptible to oxidative stress in comparison with non-cancer cells, and this phenomenon explains the higher sensitivity of cancer cells to pro-oxidants [50].

Chalcones have attracted researchers’ attention as potential anticancer compounds because of their multi-target action together with simple chemistry, as well as their low toxicity [17]. Recently, we documented excellent antiproliferative action of 1C acridine chalcone in HCT116 cells [11]. In the last decade, some articles indicated a relationship between the antiproliferative effect and induction of oxidative stress in both natural and synthetic chalcones [51,52,53,54]. This fact prompted us to evaluate the role of ROS/RNS in the antiproliferative and proapoptotic effect of chalcone 1C. 

Our results showed that co-treatment of HCT116 cells with 1C and antioxidant NAC significantly attenuated 1C-induced decrease of cancer cell survival. Because this finding indicates possible involvement of free radicals in 1C action, we further evaluated some parameters of oxidative stress. 

Firstly, we detected increased levels of ROS after 24, 48, and 72 h of treatment. Moreover, using specific indicators, we also detected higher concentrations of superoxide and nitric oxide (NO). It has been demonstrated that reaction of NO and superoxide resulted in production of peroxynitrite, a highly reactive oxidant that is probably responsible for cytotoxicity both of these oxidants [55,56]. Interaction of peroxynitrite with membrane polyunsaturated fatty acids often results in lipid peroxidation [57]. In line with this evidence, we also detected significant increase of lipid peroxides in 1C-treated cells. 

Secondly, because oxidative stress is a result of imbalance between pro-oxidants and antioxidants, we further evaluated effect of 1C on the intracellular antioxidant defence mechanism. Treatment of HCT116 with 1C significantly influenced GSH, GRH, and GPx. We found a biphasic effect of 1C on GSH levels, with a significant decrease after 24 h of incubation. This decrease of GSH levels was associated with increased activity of GR. Because GR catalyses reduction of oxidized glutathione to GSH [58], we suggest that the increase in GR activity could be an adaptive cellular mechanism to GSH depletion. Moreover, in the same time of incubation, we detected the highest increase in lipid peroxide production and increase in GPx activity. Because the role of GPx is to catalyse reduction of different peroxides including lipid peroxides [59], increase in GPx activity may be a consequence of lipid peroxide overproduction.

On the basis of the aforementioned findings, we considered the fact that increased generation of free radicals and simultaneous weak adaptive activation of endogenous antioxidant system results in oxidative stress and subsequent cell death. Our hypothesis also supports the fact that the addition of NAC significantly decreases the antiproliferative potential of 1C and increases cell survival.

In addition to lipid peroxidation, ROS can also induce DNA damage [60]. Accordingly, in 1C-treated cells, we observed significant increase of 8-oxo-7,8-dihydroguanine, the oxidation product of guanine, which is considered as a marker of oxidative damage of DNA [61]. 

Damage of DNA usually results in activation of the DNA damage response (DDR) system, which involves DNA damage detection, cell cycle arrest, and activation of repair mechanisms, followed by survival or cell death [62]. The key role in DDR is played by H2A.X, which is activated by phosphorylated ATM. Activated, that is, phosphorylated H2A.X (termed γ-H2A.X) is important for recruitment of repair proteins to DNA damage sites. Moreover, it is also considered as an indicator of DNA double-strand breaks [63]. Several genotoxic agents that cause DNA double-strand breaks, such as anthracycline antibiotics, cisplatin, or etoposide, facilitate γ-H2A.X formation. In addition, another component of DDR network phosphorylated by ATM is the SMC1 protein [64]. In the present work, we observed activation of these DDR components in 1C-treated cells indicating DNA damage. Together with the observation that 1C increased 8-oxoG levels and NAC significantly prevented either 8-oxoG production or activation of DDR markers, we suggest that 1C induced oxidative damage of DNA. To the best of our knowledge, this is the first study reporting that chalcone induces oxidative DNA damage in colorectal cancer cells. Recently, Gil and co-workers [65] presented ROS-mediated DNA damage in human lung cancer cells, which support our findings related to role of ROS in the antiproliferative effect of chalcones.

A significant amount of damage from biomacromolecules, caused by oxidants, can lead to cell death [66]. Results shown in this paper, together with our recently published data [11], suggest that chalcone 1C induces apoptosis associated with mitochondrial dysfunction. Today, it is without doubt that mitochondria play an important role in apoptosis. Permeabilization of the mitochondrial outer membrane (MOM) allows translocation of proteins localized in the intermembrane space to cytosol with subsequent caspase activation and apoptotic cell death [67]. In the present work, we found a significant increase of cytoplasmatic levels of cytochrome *c*, Smac/DIABLO, and activation of executioner caspase-3 and -7. Once cytochrome *c* is released, binds to apoptotic protease-activating factor 1 (APAF-1) in the presence of deoxyadenosine triphosphate to form the apoptosome [68] with subsequent activation of downstream effector caspases and initiation of apoptosis. In addition, Smac/DIABLO protein, by inhibiting of IAP (inhibitor of apoptosis protein) function, allows caspases to activate apoptosis [69]. Permeabilization of MOM is regulated by pro- and anti-apoptotic proteins of the Bcl-2 family. For example, it has been demonstrated that over-expression Bcl-2 blocks the release of cytochrome *c* from the mitochondria to cytosol and prevents cell death [70]. Our experiments showed increased levels of phosphorylated anti-apoptotic Bcl-2 protein, which led to loss of its function.

Apoptotic machinery can be activated in response to different stimuli, and ROS are one form of them [71]. The results presented here clearly documented the role of ROS in 1C-induced mitochondrial dysfunction and apoptosis. The addition of antioxidant NAC to HCT116 cancer cells decreased the release of cytochrome *c* and Smac/DIABLO proteins, activation of caspase-3 and -7, and PARP cleavage, as well as phosphorylation of Bcl-2 proteins. Moreover, NAC almost completely prevented 1C-induced loss of mitochondrial membrane potential. Although it is not clear if loss of MMP is an early event or consequence in apoptosis, it is broadly accepted that decrease of MMP can be associated with mitochondrial dysfunction and apoptosis as well [72].

It has been shown that MAPKs are involved in regulation of many cellular processes including cell growth, survival, and apoptosis [73].

It has been suggested that JNK phosphorylation activates apoptosis in response to different types of stress [74]. Moreover, the ability of activated JNK to release pro-apoptotic proteins such as cytochrome *c* and Smac/DIABLO has also been documented [75]. Similarly to JNK, phosphorylation of p38 also can be involved in apoptosis [76]. Moreover, although it is commonly accepted that the phosphorylation of the ERK pathway leads to cell proliferation, activation of ERK1/2 can also be involved in apoptosis [77]. Our results are consistent with aforementioned findings, as we found increased phosphorylation of all three members of MAPK signalling pathway in 1C-treated cancer cells.

Furthermore, the involvement of ROS in activation of mitogen-activated protein kinases (MAPKs) pathway has been proposed [78]. Our results support this hypothesis. As shown here, co-treatment of cells with NAC significantly decreased phosphorylation of p38 MAPK, ERK1/2, and JNK and partially suppressed activation of apoptosis machinery and decreased 1C-induced cell death. These results are in agreement with those published by Wang et al. [79], who found that new chalcone SL-4 induced apoptosis in human cancer cells through increased production of ROS and activation of the MAPK signalling pathway.

## 5. Conclusions

Our study showed that acridine chalcone 1C has an antiproliferative and pro-apoptotic effect against colorectal cancer HCT116 cells through ROS generation. Mechanistically, 1C increased ROS production with concomitant mitochondrial dysfunction associated with loss of MMP, release of cytochrome *c*, and Smac/DIABLO proteins with subsequent caspase-3 and -7 activation and apoptosis. Moreover, treatment of HCT116 cells with 1C led to the activation of the MAPK signalling pathways.

## Figures and Tables

**Figure 1 biomolecules-10-00345-f001:**
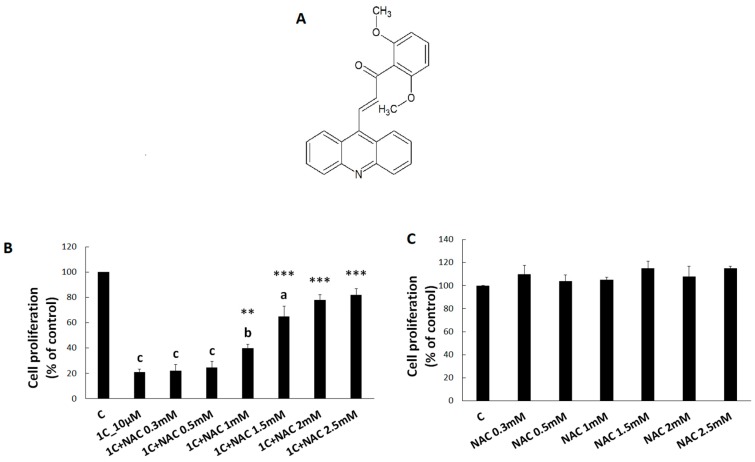
HCT116 cell proliferation treated with chalcone 1C ((2 E)-3-(acridin-9-yl)-1-(2,6-dimethoxyphenyl)prop-2-en-1-one) alone (**A**) or in combination (*N*-acetyl cysteine (NAC)/1C) (**B**) and after NAC dilutions (**C**). Data were obtained from three independent measurements. Significantly different ^a^
*p* < 0.05, ^b^
*p* < 0.01, ^c^
*p* < 0.001 vs. untreated cells (control); ** *p* < 0.01, *** *p* < 0.001 vs. 1C.

**Figure 2 biomolecules-10-00345-f002:**
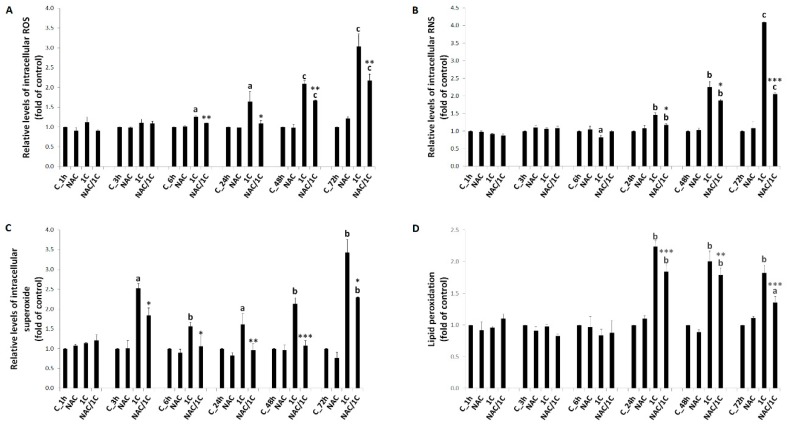
The influence of 1C and NAC/1C on free radical production in HCT116 cells. (**A**) Measurement of reactive oxygen species (ROS) levels after 6, 24, 48, and 72 h incubation (**B**) Relative levels of reactive nitrogen species (RNS) after 6, 24, 48, and 72 h incubation (**C**) Relative levels of superoxide after 3, 6, 24, 48, and 72 h incubation (**D**) Analysis of lipoperoxide production after 6, 24, 48, and 72 h incubation. Data were obtained from three independent measurements. Significantly different ^a^
*p* < 0.05, ^b^
*p* < 0.01, ^c^
*p* < 0.001 vs. untreated cells (control); * *p* < 0.05, ** *p* < 0.01, *** *p* < 0.001 vs. 1C.

**Figure 3 biomolecules-10-00345-f003:**
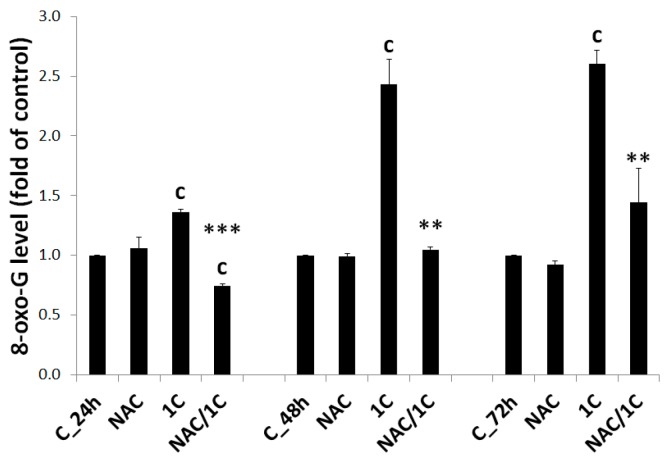
Oxidative DNA damage detection using 8-oxo-7,8-dihydroguanine (8-oxoG) level measurement after 24, 48, and 72 h incubation of HCT116 cells with 1C and NAC/1C. Significantly different ^c^
*p* < 0.001 vs. untreated cells (control); ** *p* < 0.01, *** *p* < 0.001 vs. 1C.

**Figure 4 biomolecules-10-00345-f004:**
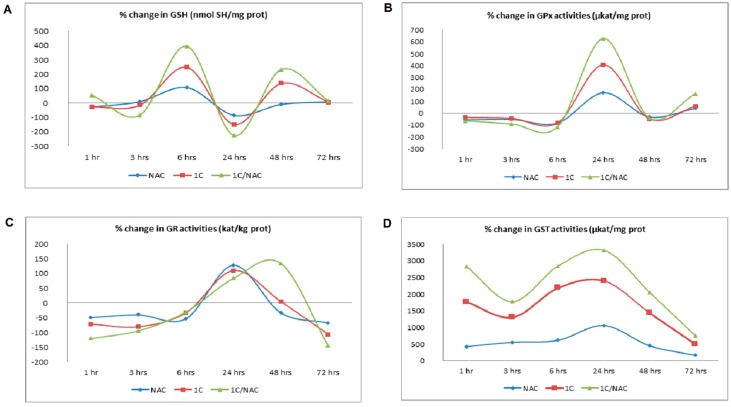
Antioxidant status of HCT116 cells after 1C, NAC, and NAC/1C treatment. The influence of 1C and combination of NAC/1C on glutathione content (**A**), glutathione peroxidase (GPx) activity (**B**), glutathione reductase (GR) activity (**C**), and glutathione-S-transferase (GST) activity (**D**). The experiments were performed in triplicate and measured parameters were calculated per milligram or kilogram of protein vs. untreated cells (control).

**Figure 5 biomolecules-10-00345-f005:**
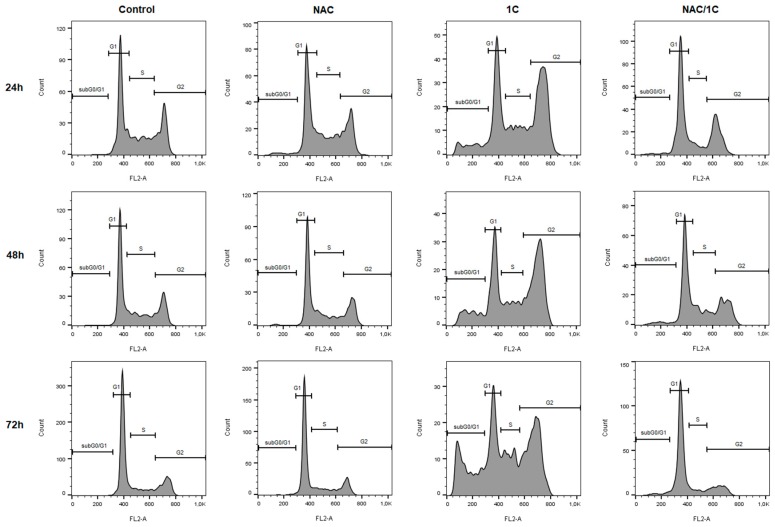
Representative histograms of cell cycle distribution in HCT116 cells treated with NAC, 1C, or NAC/1C after 24, 48, and 72 h.

**Figure 6 biomolecules-10-00345-f006:**
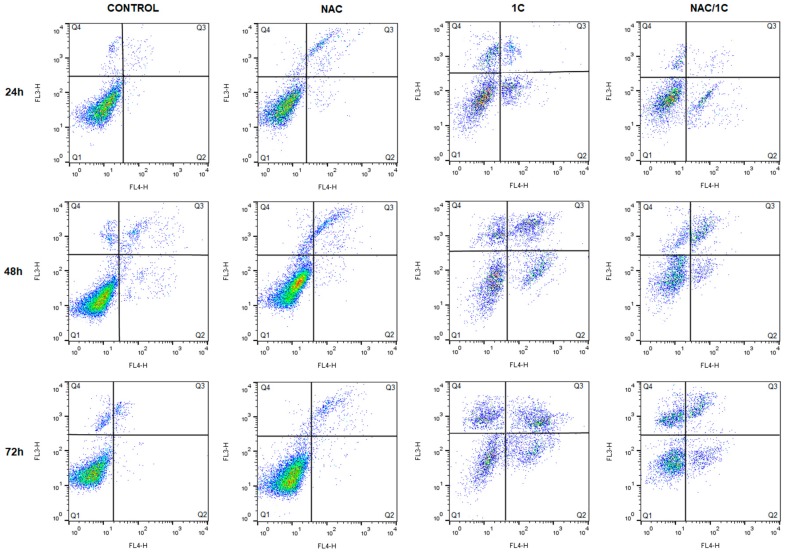
Representative dot-blot diagrams of annexin V/PI staining in HCT116 cells after treatment with NAC, 1C, and NAC/1C.

**Figure 7 biomolecules-10-00345-f007:**
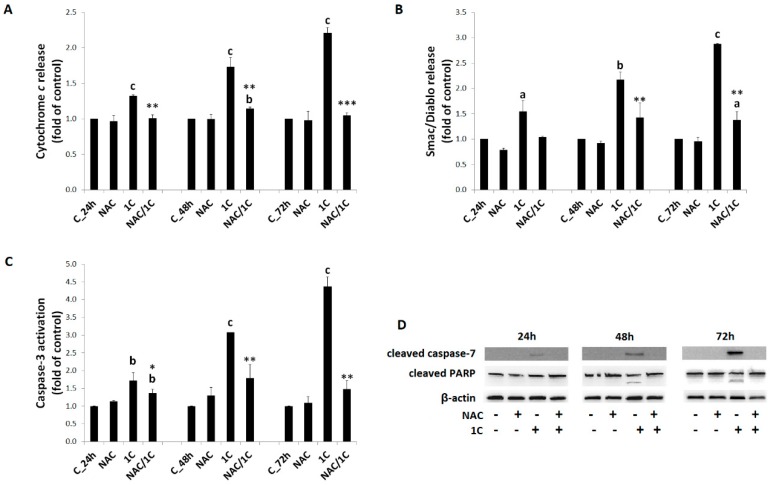
Mitochondrial apoptotic pathway alteration in 1C-treated and NAC/1C-co-treated HCT116 cells represented by cytochrome *c* release (**A**), expression of Smac/DIABLO (**B**), and caspase-3 (**C**) or caspase-7 activation and PARP cleavage (**D**). Significantly different ^a^
*p* < 0.05, ^b^
*p* < 0.01, ^c^
*p* < 0.001 vs. untreated cells (control); * *p* < 0.05, ** *p* < 0.01, *** *p* < 0.001 vs. 1C.

**Figure 8 biomolecules-10-00345-f008:**
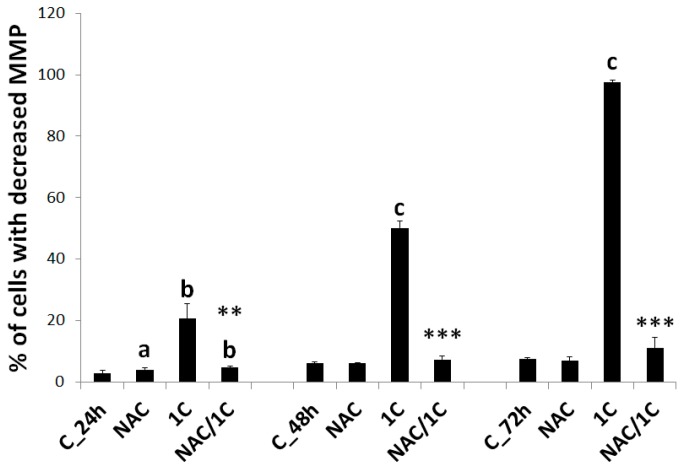
Mitochondrial membrane potential changes after 1C or NAC/1C treatment. Significantly different ^b^
*p* < 0.01, ^c^
*p* < 0.001 vs. untreated cells (control); ** *p* < 0.01, *** *p* < 0.001 vs. 1C.

**Figure 9 biomolecules-10-00345-f009:**
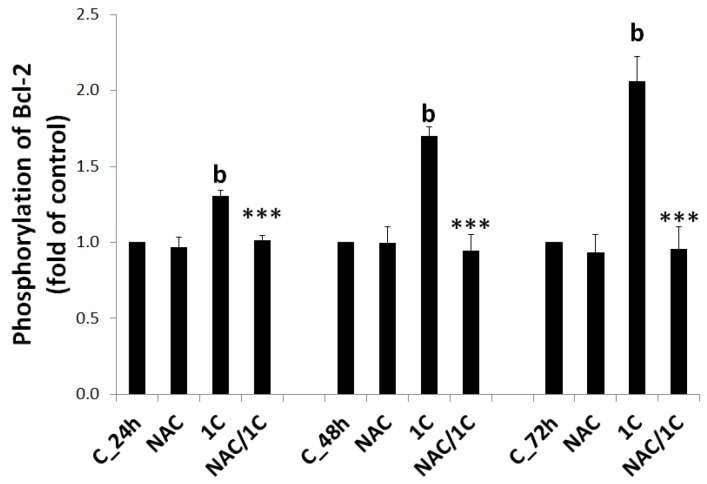
Relative levels of phosphorylated bcl-2 after 24, 48, and 72 h of 1C or NAC/1C treatment. Significantly different ^b^
*p* < 0.01 vs. untreated cells (control); *** *p* < 0.001 vs. 1C.

**Figure 10 biomolecules-10-00345-f010:**
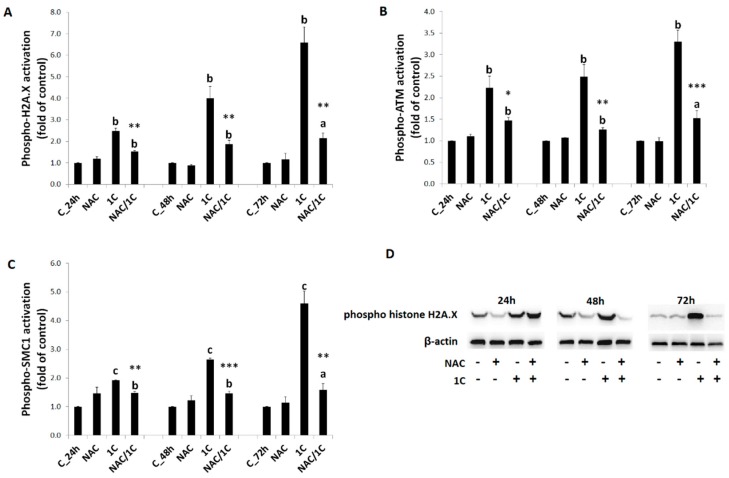
Analysis of DNA damage-related proteins. Phospho histone H2A.X (**A**,**D**), phospho-ataxia telangiectasia mutated kinase (ATM), (**B**) and phospho-structural maintenance of chromosomes 1 (SMC1) (**C**) levels after 24, 48, and 72 h of 1C or NAC/1C treatment. Significantly different ^a^
*p* < 0.05, ^b^
*p* < 0.01, ^c^
*p* < 0.001 vs. untreated cells (control); * *p* < 0.05, ** *p* < 0.01, *** *p* < 0.001 vs. 1C.

**Figure 11 biomolecules-10-00345-f011:**
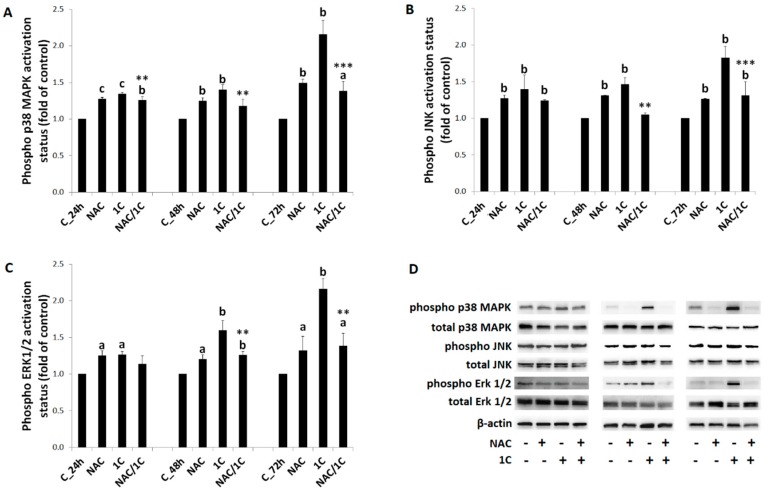
Phosphorylation status of mitogen-activated protein kinase (MAPK) proteins. Phosphorylation changes of p38 MAPK (**A**,**D**), JNK (**B**,**D**), and ERK1/2 (**C**,**D**) as the result of 1C or NAC/1C combination treatment. Significantly different ^a^
*p* < 0.05, ^b^
*p* < 0.01, ^c^
*p* < 0.001 vs. untreated cells (control); ** *p* < 0.01, *** *p* < 0.001 vs. 1C.

**Table 1 biomolecules-10-00345-t001:** Flow cytometry staining.

Analysis	Staining Solution	Manufacturer
ROS	DHR123 (Dihydrorhodamine 123), final concentration 200 nM	Sigma-Aldrich, St. Louis, MO, USA
RNS	DAF-FM (Diaminofluorescein-FM) diacetate, final concentration 2 mM	Sigma-Aldrich, St. Louis, MO, USA
Lipid peroxidation	BODIPY 581/591 C11, final concentration 1 mM	Sigma-Aldrich, St. Louis, MO, USA
Superoxide anion	MitoSox Red mitochondrial superoxide indicator, final concentration 5 µM	Sigma-Aldrich, St. Louis, MO, USA
Externalization of phosphatidylserine	Annexin V-FITC, 1:100 Propidium iodide, final concentration 25 µg/mL	BD Biosciences Pharmingen, San Diego, CA, USA
Caspase-3 activation	Cleaved caspase-3-PE, 1:200	BD Biosciences Pharmingen, San Diego, CA, USA
Cytochrome *c* release	Cytochrome *c* antibody (6H2) FITC conjugate, 1:200	Invitrogen, Carlsbad, CA, USA
Smac/DIABLO release	Smac/DIABLO rabbit mAb, 1:200	Cell Signaling Technology, Danvers, MA, USA
	Goat anti-rabbit IgG (H + L) secondary antibody, Alexa Fluor 488, 1:500	Thermo Scientific, Rockford, IL, USA
Mitochondrial membrane potential	TMRE (tetramethylrhodamine ethyl ester perchlorate), final concentration 0.1 µM	Sigma-Aldrich, St. Louis, MO, USA
Protein analysis	Phospho-Bcl-2 (Ser70) rabbit mAb Alexa Fluor 488 conjugate, 1:200 Phospho-p44/42 MAPK (Erk1/2) (Thr202/Tyr204) (E10) mouse mAb, 1:2000 Phospho-SAPK/JNK (Thr183/Tyr185) (G9) mouse mAb (PE conjugate), 1:200 Phospho-p38 MAPK (Thr180/Tyr182) (3D7) rabbit mAb PE conjugate, 1:200	Cell Signaling Technology, Danvers, MA, USA
	Goat anti-rabbit IgG (H+L) secondary antibody, Alexa Fluor 488, 1:500	Thermo Scientific, Rockford, IL, USA
DNA damage	Anti-pATM, PE conjugated antibody, 1:200 Anti-pHistone H2A.X, PerCP conjugated antibody, 1:200 Anti-pSMC1, Alexa Fluor 488 Antibody, 1:200	Millipore Corporation, Temecula, CA, USA
	Anti-oxoguanine 8 antibody	Abcam, Cambridge, United Kingdom
	Goat anti-mouse IgG (H + L) secondary antibody, Alexa Fluor 488	Thermo Scientific, Rockford, IL, USA

**Table 2 biomolecules-10-00345-t002:** List of Western blot antibodies.

**Primary Antibodies**	**Mr (kDa)**	**Origin**	**Manufacturer**
β-actin	45	Mouse	Cell Signaling Technology, Danvers, MA, USA
p38 MAPK	43	Rabbit	
Phospho-p38MAPK	43	Rabbit	
p44/42 MAPK (Erk1/2)	42 + 44	Rabbit	
Phospho-p44/42 MAPK (Erk 1/2)	42 + 44	Mouse	
JNK1	48	Mouse	
Phospho-SAPK/JNK	54	Mouse	
Cleaved caspase-7	18	Rabbit	
PARP	116 + 89	Rabbit	
Phospho-histone H2A.X	15	Rabbit	
**Secondary Antibodies**	**Mr (kDa)**	**Origin**	**Manufacturer**
Anti-rabbit IgG HRP	-	Goat	Cell Signalling Technology, Danvers, Massachusetts, USA
Anti-mouse IgG/HRP	-	Goat	Dako, Glostrup, Denmark

JNK1- c-Jun N-terminal kinase; SAPK- Stress-activated protein kinase; PARP- Poly (ADP-ribose) polymerase.

**Table 3 biomolecules-10-00345-t003:** Cell cycle analysis of HCT116 cells after 24, 48 and 72 h incubation with NAC, 1C, or NAC/1C.

	SubG_0_/G_1_	G_1_	S	G_2_
**C_24h**	1.77 ± 0.74	44.65 ± 0.65	28.45 ± 2.55	25.25 ± 2.55
**NAC**	2.71 ± 1.36	41.30 ± 2.80	25.55 ± 0.45	30.60 ± 3.50
**1C**	11.30 ± 2.31 ^b^	29.25 ± 3.15 ^b^	14.40 ± 2.10 ^b^	45.25 ± 1.45 ^c^
**NAC/1C**	4.25 ± 1.23 ^a^**	50.75 ± 3.45 ***	15.85 ± 4.45 ^b^	29.35 ± 2.35 **
**C_48h**	0.95 ± 0.65	55.45 ± 6.45	19.10 ± 6.50	24.30 ± 0.90
**NAC**	1.92 ± 0.38	47.95 ± 0.15	23.60 ± 2.00	26.55 ± 2.55
**1C**	11.85 ± 0.55 ^b^	31.80 ± 2.20 ^b^	17.75 ± 2.65	38.35 ± 1.25 ^b^
**NAC/1C**	5.55 ± 0.95 ^a^***	52.95 ± 0.35 **	16.85 ± 3.85	24.50 ± 3.20 **
**C_72h**	0.71 ± 0.03	68.30 ± 6.00	13.25 ± 1.95	17.85 ± 4.45
**NAC**	1.53 ± 0.77	68.95 ± 6.35	14.75 ± 2.55	14.55 ± 3.15
**1C**	15.63 ± 7.18 ^b^	26.25 ± 1.95 ^c^	18.80 ± 1.00	39.00 ± 4.20 ^b^
**NAC/1C**	3.85 ± 1.24 ^a^*	67.75 ± 0.65 ***	10.65 ± 1.65 **	17.75 ± 0.25 **

The results are presented from three independent measurements as the mean ± standard deviation (SD). Significantly different ^a^
*p* < 0.05, ^b^
*p* < 0.01, ^c^
*p* < 0.001 vs. untreated cells (control); * *p* < 0.05, ** *p* < 0.01, *** *p* < 0.001 vs. 1C.

**Table 4 biomolecules-10-00345-t004:** Externalization of phosphatidylserine after treatment with NAC, 1C, and NAC/1C.

	Live (Q1)	Early Apoptotic (Q2)	Late Apoptotic (Q3)	Death (Q4)
**C_24h**	97.80 ± 0.65	0.05 ± 0.04	0.18 ± 0.10	2.00 ± 0.49
**NAC**	90.95 ± 1.67	1.61 ± 1.14 ^b^	6.20 ± 3.43 ^b^	1.23 ± 0.633 ^b^
**1C**	68.50 ± 0.98 ^c^	17.84 ± 2.25 ^c^	3.99 ± 0.99 ^c^	9.78 ± 0.34 ^c^
**NAC/1C**	86.65 ± 1.59 ^b^***	10.92 ± 2.35 ^c^***	0.71 ± 0.32 ^b^**	1.75 ± 0.45 ***
**C_48h**	93.55 ± 0.86	1.37 ± 0.53	2.28 ± 0.62	2.75 ± 2.00
**NAC**	91.50 ± 0.98	0.16 ± 0.13	8.10 ± 0.65 ^b^	0.27 ± 0.19 ^b^
**1C**	42.55 ± 5.76 ^c^	16.53 ± 2.96 ^c^	17.23 ± 5.69 ^c^	23.70 ± 3.02 ^c^
**NAC/1C**	62.15 ± 1.84 ^b^**	10.49 ± 0.35 ^c^*	15.61 ± 3.66 ^c^	11.75 ± 1.43 ^b^**
**C_72h**	94.30 ± 0.41	0.78 ± 0.21	1.38 ± 0.51	3.56 ± 0.29
**NAC**	94.95 ± 0.20	0.06 ± 0.05 ^c^	4.85 ± 0.05	0.17 ± 0.09
**1C**	31.85 ± 5.59 ^c^	11.29 ± 0.58 ^c^	25.71 ± 4.07 ^c^	31.20 ± 0.89 ^c^
**NAC/1C**	61.10 ± 1.23 ^b^**	8.83 ± 1.48 ^c^*	11.83 ± 0.96 ^b^*	18.24 ± 0.70 ^b^***

The results are presented from three independent measurements as the mean ± standard deviation (SD). Significantly different ^b^
*p* < 0.01, ^c^
*p* < 0.001 vs. untreated cells (control); * *p* < 0.05, ** *p* < 0.01, *** *p* < 0.001 vs. 1C.

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
