# Peer review of "Antiproliferative Effect of Acridine Chalcone Is Mediated by Induction of Oxidative Stress"

_biomolecules, 2020, doi:10.3390/biom10020345_

Round 1

Reviewer 1 Report

The revised manuscript by Takac et al. proposes that the anti-proliferative and pro-apoptotic action of Acridine Chalcone is caused by oxidative stress.

Unfortunately, the authors have failed to address several major concerns that would have strengthened the manuscript.

Although the manuscript now reads 0.02% DMSO, rather than <0.2% DMSO as stated in the original submission, the controls should always be included at the time of the experiment. In this case, the methods section clearly says that the compounds were dissolved in DMSO yet data corresponding to the vehicle control is still missing.

Further, the authors cite a previous study in which they used BrdU staining (Takac et al., 2018) but this does not change the inherent limitations of MTT assays  presented in this study.  If MTT assays are used then they should be interpreted correctly with precise language.

Author Response

Although the manuscript now reads 0.02% DMSO, rather than <0.2% DMSO as stated in the original submission, the controls should always be included at the time of the experiment. In this case, the methods section clearly says that the compounds were dissolved in DMSO yet data corresponding to the vehicle control is still missing.

Answer: As we stated previously, we used absolutly non-toxic (and even with no measurable effect) concentration of DMSO (more than 20 years of experience). According reviewer we should repeat almost all experiments and we would need a few months to do it which is not possible for us now.

Further, the authors cite a previous study in which they used BrdU staining (Takac et al., 2018) but this does not change the inherent limitations of MTT assays  presented in this study.  If MTT assays are used then they should be interpreted correctly with precise language.

Answer: We changed interpretation of MTT test

Reviewer 2 Report

This manuscript describes the anticancer effect of chalcone 1C by potential increase of ROS. And evidences of 1C-drived ROS cell death was shown by treatment with NAC. The approach for this search is appropriately designed and well supported. However, there are some issues to be published.

Authors need a careful English editing, there are many grammatical errors. Chemical structure must be shown in the Figure. Antibody list must be in the text of “material and method” not as in the Table. Also, all materials for FACS analysis must be the text. Labelling for all figure is distractive and too small to see.

Author Response

Authors need a careful English editing, there are many grammatical errors.

Answer: If the manuscript will be accepted, we will used English editind service of the publisher.

Chemical structure must be shown in the Figure.

Answer: added in Fig. 1

Antibody list must be in the text of “material and method” not as in the Table. Also, all materials for FACS analysis must be the text.

Answer: We tried to meet this requirement. However, when the list of antibody (and other material for flow cytometry) was in the text, it was difficult to read it. In the table form, it is more clear  and we prefer this form.

Labelling for all figure is distractive and too small to see.

Answer: Corrected

Reviewer 3 Report

Review on Manuscript entitle “Antiproliferative effect of acridine chalcone is  mediated by induction of oxidative stress” authored by P Takat et al.

In this article, authors evaluated the contribution of oxidative stress in the antiproliferative effect of acridine chalcone 1C in human colorectal HCT116 cells. They demonstrated that chalcone 1C increases ROS/RNS and superoxide production with reduced cellular antioxidant defence mechanism, which promotes mitochondrial dysfunction, DNA damage and apoptosis. Importantly, N-acetyl cysteine (NAC) treatment significantly attenuated all the effects of acridine chalcone 1C in colorectal cancer cells. These results suggest a potential use of chalcone as a promising anti‑cancer agent for colorectal cancer.

The Manuscript is well written, clear and straightforward. By mean of an ample set of experiments (Flow cytometry analysis, measurement of antioxidant activities, GSH content, DNA damage, cell cycle and apoptosis and caspase activity) authors demonstrate a pro-oxidant effect of acridine chalcone in HCT116  cells to induce apoptosis.

This is an interesting paper with a clear translational potential. Nevertheless, there are some major comments which should be addressed before its publication.

Main comments:

Conclusion are based on results on only one cell line HCT116 cells. At least, main results should be repeated on a second CRC cell line (NAC recue of apoptosis).

In addition, the effect of acridine chalcone in a non tumoral cell line should be included (non transformed colon cancer cell line), at least by mean of the analysis on cell viability (MTT assay).

Minor comments:

Resolution of Figure 2 and Figures 10 and 11 should be augmented.

Author Response

Conclusion are based on results on only one cell line HCT116 cells. At least, main results should be repeated on a second CRC cell line (NAC recue of apoptosis).

Answer: In 2018, we published an article „New chalcone derivative exhibits antiproliferative potential by inducing G2/M cell cycle arrest, mitochondrial-mediated apoptosis and modulation of MAPK signalling pathway“ by Takac et al. In this article, we studied effect of 7 newly synthesized chalcone derivatives on several cancer cell lines. Chalcone 1C was selected as the most active compound against HCT116 cells and all experiments were provided with chalcone 1C and HCT116 cells.

Because some studies indicate that antiproliferative effect of chalcones may be associated with induction of oxidative stress, in the present paper we continued in our experiments and our attention was focused on the role of reactive oxygen/nitrogen species in antiproliferative effect of chalcone 1C in HCT116 cells.

In addition, the effect of acridine chalcone in a non tumoral cell line should be included (non transformed colon cancer cell line), at least by mean of the analysis on cell viability (MTT assay).

Answer: Effect of acridine chalcone 1C on non-cancer cells were published in our previous article (Takac et al., 2018). However, because we have not non-cancer colorectal cells, we used either fibroblast (3T3) or mammary cells (MCF-10A). As we documented, in both cell lines LD50 was several times higher when compared with LD50 for HCT116 cells

Minor comments:

Resolution of Figure 2 and Figures 10 and 11 should be augmented.

Answer: adjusted

Round 2

Reviewer 3 Report

Thank you for the new version.

As indicated by the authors, previous work, "New chalcone derivative exhibits antiproliferative potential by inducing G2/M cell cycle arrest, mitochondrial-mediated apoptosis and modulation of MAPK signalling pathway“ by Takac et al. there was studied the effect of 7 newly synthesized chalcone derivatives on several cancer cell lines, being Chalcone 1C the most active compound against HCT116 cells. Authors deep into the antiproliferative effect of chalcones  which may be associated with induction of oxidative stress. Conclusions are based on the results in this cell line. Unfortunately, I still believe that conclusions will be strengh if validated in a second cell line, where chalcone 1C also induces oxidative stress. I would encourage authors to validate the main results in a second cell line where chalcone 1C exert a similar antiproliferative effect.

Author Response

We aggree with reviewer that conclusions may be strengh if validated in an another cancer cell line. However, as we mentioned previously, this work is continuation of our previous work (Takac et al., 2018) at what we used several cancer and non-cancer cell lines but only for MTT-based screening. On the base of this, we selected chalcone 1C in HCT116 cells and we studied mechanism of its antiproliferative effect. In this study we used nearby 30 different assays. In the present manuscript we wanted only to profound our knowledge about the potential mechanism of 1C in HCT116 cells and we studie the role of oxidative stress in it. Additional experiment on different cancer cell line would give some information about the role of oxidative stress in the mechanism of antiproliferative effect of chalcone 1C in general. On the other hand, it would not give us more information about this effect in HCT116 cells, which was our goal.